# Exploring user experience: A qualitative analysis of the use of a physical activity support app for people with heart failure

Andreas Blomqvist[1]*, Anna Strömberg[1,2], Marie Lundberg[1], Maria Bäck[1,3], Tiny Jaarsma[1], Leonie Klompstra[1]

1 Department of Health, Medicine and Caring Sciences, Linköping University, Linköping, Sweden,
2 Department of Cardiology, Linköping University, Linköping, Sweden, 3 Department of Occupational Therapy and Physiotherapy, Sahlgrenska University Hospital, Gothenburg, Sweden

* andreas.blomqvist@liu.se

## Abstract

### Introduction

Heart failure (HF) is a prevalent and debilitating global health issue, affecting approximately 65 million people worldwide. Physical activity is recommended for HF management, yet many people with HF remain sedentary. This study explored user experiences with an mHealth tool called the Activity Coach (the app), designed to support physical activity among people with HF.

### Purpose

The purpose of this study was to describe users' experiences with an app designed for supporting physical activity in people with HF.

### Methods

Using a qualitative design, semi-structured telephone interviews were conducted with ten people with HF who used the app for 12 weeks. Thematic analysis was used to analyse the data. The interviews were transcribed verbatim, data items identified and coded and themes were generated based on these codes. The themes were subsequently defined and described.

### Results

The analysis yielded two themes: "Cultivating awareness of physical activity engagement" and "Motivation through enjoyment in the monitoring process and through physical and emotional changes". We found that while the app was found easy to use, users faced challenges in defining and tracking physical activity. The app increased users' awareness and motivation for physical activity and helped establish new routines. Users also experienced improved physical health and emotional well-being.

**Data availability statement:** All data is uploaded as a zip-file. It contains the verbatim transcripts from all study participants.

**Funding:** The author(s) received no specific funding for this work.

**Competing interests:** I have read the journal's policy and the authors of this manuscript have the following competing interests: The author Andreas Blomqvist, PhD-student at Linköping University, is an employee and shareholder of the company that developed the app and Optilogg (CareLigo AB, Sweden). This does not alter our adherence to PLOS ONE policies on sharing data and materials.

## Conclusions

The app created physical activity awareness and motivation, and the study shows that mHealth may be used to increase physical activity motivation and engagement in people with heart failure.

## Introduction

Heart failure (HF) is a global problem, affecting close to 65 million people worldwide [1]. HF is a debilitating disease, characterized by recurring re-admissions and symptoms such as fatigue, shortness of breath and oedema [2]. In addition to pharmacological therapy and multidisciplinary care, current treatments guidelines from the European Society of Cardiology recommend physical activity for people with HF [2], as it significantly improves prognosis [3–8]. Despite these recommendations, many people with HF lead sedentary lifestyles [9,10] and could thus benefit from increased physical activity. There are several barriers to increased physical activity for this population, and one important such barrier is fear of movement, which is a result of trying to avoid discomfort and anxiety from symptoms [11]. Other barriers identified for this population are symptoms, comorbidities, extreme weather or seasonal changes, costs of leisure activities, and transportation [12].

Adequate self-care behaviour is important in order to improve quality of life, physical function and decrease symptoms for people with HF [13], and physical activity is one aspect of self-care that is attractive to influence since the effects on prognosis are profound. However, behaviour change is difficult to achieve and even more difficult to maintain [14]. Modern technology, such as mHealth (or "medical health practice using mobile devices" as defined by the WHO) [15], can play an important role in supporting behaviour change. The portable nature in itself and constant availability can address several previously mentioned barriers, e.g., as the user will not have to leave home. For mHealth tools to be effective, focus on usability is important [16–18], and assessment of usability for elderly users with chronic conditions, like people with HF often are [1], may need other considerations compared to a younger population [19]. In spite of a large number of mHealth services or products having entered the market in recent years, there are still knowledge gaps related to user experience [20] and qualitative data should be included in such analyses [21]. While there is a wealth of publications on interventions for exercise in the HF population, there are few, if any, mHealth interventions aimed at supporting non-exercise physical activity [22] in the HF population. Such an intervention has been developed [23], and in this paper we explore the user experiences associated with using this tool.

### Purpose

The purpose of this study was to describe users' experiences with an mHealth-tool designed for supporting physical activity in people with HF.

## Materials and methods

### Design

This was a qualitative descriptive study, analysed with thematic analysis [24,25]. The writing of the manuscript was guided by the Standards for Reporting Qualitative Research (SRQR) [26].

### The mHealth tool

An mHealth-tool called Optilogg is currently used in several health care regions in Sweden to support people with HF to improve self-care behaviour through symptom monitoring, a flexible diuretics regimen and an interactive education module, with high system adherence [27,28]. On the Optilogg platform, a novel feature called the Activity Coach was developed to support one specific self-care behaviour, namely physical activity [23]. The Activity Coach will throughout this article be referred to as the "app". The Optilogg home screen and the Activity Coach app are shown in Fig 1.

When a person with HF gets the app activated on his/her Optilogg device, an introductory week starts where the user will receive education on physical activity in the context of HF. The purpose of this education is to first establish personal perceptions that facilitate behaviour change. The education consists of one or two sentences written using layman terms, with the purpose of establishing positive outcome expectancy, self-efficacy and goal congruence [23]. One example from that introductory week is "Everyone can be physically active; it's just about choosing what feels right for you! It can be a walk around the house, standing up now and then in front of the TV, or doing household chores.". If the user wants to, they can read an extended version of the same information of about 75–100 words.

Once that first week is completed, the user is encouraged to daily register physical activity in increments of 10 minutes (regardless of the specific activity undertaken or the intensity with which it was performed) in the graphical user interface (GUI, see Fig 1). In this manuscript we refer to that procedure as "tracking" physical activity. There were information readily available listing examples of what constitutes physical activity in the GUI. Examples from that list include going for a walk, gardening, sit-to-stand transitions, seated rubber band exercises, leg kicks or vacuuming the house.

At the end of each week a summary screen appears, which informs the user of the total tracked physical activity during the previous week. There is also an opportunity to set a goal for the upcoming week. The goal for next week was a choice between "less than the previous week", "same as the previous week" and "more than the previous week". If a goal was already set for the current week, feedback on whether that goal was reached or not is provided.

The user could always check past registered activity and trends, as well as track the progress to achieving the current goal in a part of the "History tab". The app and the entire Optilogg system is an Android-implementation on a CE-marked 7" tablet computer.

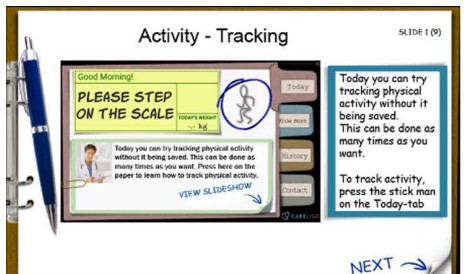 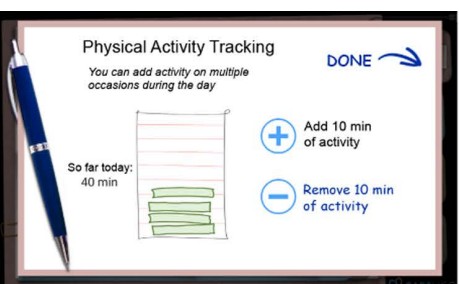 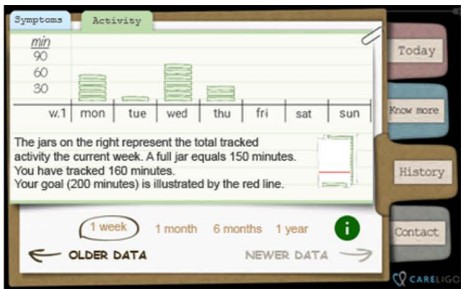

**Fig 1. The first panel shows the first day the stick figure appears with the instructions to test tracking physical activity.** The middle panel shows where the user tracks physical activity in increments of ten minutes. The right-most panel shows the history tab with the weekly goal being illustrated by the red line on the "jar".

## Ethical approval and population

The app was tested in a pilot randomized controlled clinical trial (ClinicalTrials.gov identifier: NCT05235763), and the duration of the study was 12 weeks [29]. This study was conducted in accordance with the principles of the Declaration of Helsinki and the ethics approval for this study was obtained from Sweden's ethical review board on 2023-02-19 (Dnr 2023-00799-02), and written informed consent was obtained from all individual participants included in the study. All app users from the study who were still alive (n = 9) were invited to participate. One person who was randomized to the control group but used the app 12 weeks after the study was also invited to participate. It was assumed that between 6 and 12 interviews [30], would give sufficient data to adequately achieve the purpose of the study. All users were listed at primary health care centres in the same region in Sweden and recruited between 2023-05-05 and 2023-08-31.

## Data collection

The semi-structured interviews were conducted via mobile phone by authors AB (n = 3) and ML (n = 7)). AB collected the data within the pilot RCT study, and ML is a critical care nurse with a master's degree working as a study co-ordinator. The majority of the interviews were conducted by MB, the verbatim transcripts were made by a third party, and all other researchers (LK, TJ, MB and AS) took part in the data analysis.

In total ten interviews were conducted between 2023-05-05 and 2023-08-31. The interview guide was developed by the authors, with inspiration from other semi-structured interview studies aimed at exploring user experiences of apps [31,32], see Table 1.

The interviews were recorded and subsequently transcribed verbatim. The mean time elapsed from when the study participant had stopped using the app until the interview was conducted was 38 days. The length of the interviews ranged from 15 to 32 minutes, and the median length was 25 minutes (inter quartile range 22–29 minutes). The background characteristics of the users are listed in Table 2. The users had a median adherence to the app of 69%, defined as the number of days they had tracked physical activity divided by the number of days in the study (see Table 2). The weekly adherence was 88%. In the intervention study in which the users were originally recruited, they were labelled as physically inactive or active [29], based on a single self-report item previously used to identify physically inactive people with HF [33].

## Analysis

In this study we used a descriptive qualitative approach, relying on inductive thematic analysis as described by Braun and Clarke and the listed six phases of reflexive thematic analysis [24,25]. The six phases are familiarization with the data,

**Table 1. The interview guide (translated from Swedish).**

| Question |
| --- |
| What does physical activity mean for you? |
| What made you want to participate in this research project? |
| Can you tell me about your experiences using the activity coach? |
| How did you experience registering physical activity via the "stick figure"? |
| Every week, your activity was summarized on the screen, and you could set a goal for the coming week. How did you experience that? |
| You can also use the History tab to view how you registered activity on previous days and weeks. Can you tell me about your experience with that? |
| Did you use the activity coach in ways other than those we have discussed? |
| How much did you use the activity coach? |
| Do you see anything that should be developed with the activity coach to make it better? |
| If you were offered to continue using the activity coach, how would you view that? |
| Is there anything else you would like to reflect on, highlight, or talk about? |

**Table 2. Background characteristics of the users.**

| n = 10 | |
|---|---|
| Sex | |
| Male | 4 |
| Female | 6 |
| Age (median [Q1-Q3]) | 80 [72-87] |
| Single-person household | 5 |
| NYHA-class | |
| II | 5 |
| III | 5 |
| HF type | |
| HFrEF | 3 |
| HFmrEF | 4 |
| HFpEF | 3 |
| Diabetes | 2 |
| Hypertension | 5 |
| Kidney disease | 4 |
| Atrial fibrillation | 6 |
| Physically inactive | 8 |
| App adherence (median [Q1-Q3]) | 69% [24% - 97%] |

*Q1-Q3 – inter quartile range, NYHA – New York Heart Association, HF – heart failure, HFrEF – HF with reduced ejection-fraction, HFmrEF – HF with moderately reduced ejection-fraction, HFpEF – HF with preserved ejection-fraction, KCCQ – Kansas City cardiomyopathy questionnaire.*

generating initial codes, generating themes, reviewing and developing themes, defining and naming themes and writing up the report.

The verbatim transcripts were made by an independent person and were subsequently checked against the audio recordings for accuracy by AB and notes were taken to record early impressions. AB and LK read the transcribed interviews to get to know the data. The authors AB and LK separately familiarized themselves with the data, followed by a work meeting where these impressions were discussed. All interviews were analysed once more following this working session, and this concluded the first phase of the thematic analysis.

Four interviews were selected to identify each data item. The method of identifying data items was compared and AB and LK then proceeded to identify the remaining data items in all interviews, where information carrying sentences were marked. A code list was then produced, first individually (AB and LK) and then discussed further in group session (AB, LK and AS) where codes and reasoning were compared and discussed. The code list was adapted to capture the actual words recorded but also information not explicitly mentioned (semantic and conceptual reading) when coding the data. The coding process ultimately generated 33 codes, and this concluded the second phase of the analysis.

The third phase is where all the codes are analysed for patterns, or similarities, i.e., searching to find codes united around a core concept and was carried out by AB and LK. They also produced a first set of themes and sub-themes to enter the fourth phase. AB worked on the themes to re-group and re-label as to better try to capture meaningful patterns. In the bridge between phases four and five, AB and ML went back to the original transcripts and read them through the lens of the generated themes and sub-themes and made adjustments. Changes happened on all levels, as some new codes were generated and others removed or merged (new tally was 31 codes), thus requiring re-coding of the interview transcripts, but then also on the grouping of codes into themes as well as changing the themes themselves. A first draft of tentative themes and sub-themes was developed by AB and reviewed and revised by the other authors (AS, TJ, ML and MB). The initial draft, themes and sub-themes had some overlap and sub-themes were very close to the codes and were

therefore revised. The final analysis resulted in two themes and no sub-themes level. Lastly, all authors contributed to producing the report of results, including the discussion. Throughout the report, quotes included will be labelled with age and sex, as those are factors that influence user experience and preferences about mHealth [34–37].

## Results

The analysis resulted in two themes: (1) "Cultivating awareness of physical activity engagement" and (2) "Motivation through enjoyment in the monitoring process and through physical and emotional changes".

### Theme 1 – Cultivating awareness of physical activity engagement

This theme revolves around users' increased awareness and understanding of physical activity, and experiences revolved around daily tracking of physical activity in the app, heightened awareness of what physical activity entails, their own physical activity levels, and ideas on how the app could be used better. Although an introduction of what physical activity is and can be, during the first week of using the app, users did experience some challenges in identifying which activities that constitute physical activity, making it challenging to know which activities to track. In addition to experiencing difficulties in knowing what to track in the app, users also noted challenges in accurately recalling their activities from earlier in the day when tracking physical activity, in particular how much physical activity that they had done.

*It's these small things you do like vacuuming or cleaning the windows, and things like that, then you don't really think about it or how much time you spent.*

*-Woman, 71 years*

Despite recall and definition issues, many users found the app easy to navigate and use, as explained below:

*Yes, it was just about keeping track of, for example, when I went out and how long I was out, and then I entered it when I came back in, how many minutes I had been out walking, and there were no difficulties with that at all.*

*-Woman, 77 years*

-others encountered challenges such as difficulty inputting data as desired (e.g., preference for hours being tracked instead of minutes). Similarly, when the user after each week gets a chance to set a goal for the upcoming week there was a negative experience relating to lack of flexibility when specifying the goal.

In addition to the earlier mentioned recall issues, some users simply forgot to track a certain day and then it was not possible to go back to previous days and track retrospectively, which was frustrating to some.

Users described that awareness of physical activity could create enjoyment related to feelings of learning something new about oneself, relating to one's physical activity. What was learned could be either good or bad, but the learning experience was positive regardless.

*The positive thing was that I had never really thought about it. It's like you realize just how little you actually do, and that has been a positive experience.*

*-Man, 61 years*

Where some users would become increasingly aware about how sedentary their lifestyle was by monitoring their physical activity, others would become aware that they were adequately physically active. The users belonging to the latter

category believed the app would be more beneficial for people who are less physically active, suggesting the app did not meet their current needs.

*I don't believe it would help me, but maybe it would if you were much less mobile and hardly move at all. I think it would be better for them.*

*-Woman, 71years*

Users experienced increased understanding about what kinds of physical activity that could be appropriate for them, given their situation as someone suffering from HF, and learning this and performing the aforementioned activity seemed to increase self-efficacy.

*Yeah, well, it's just some gentle exercises I can do by myself, like lifting my legs and arms. That's pretty much all there is to it. Also, some squats and getting up and down from the chair. I thought it was very unpleasant when I got palpitations and things like that, but when I felt that I started to move more, it all calmed down in some way, and then I became bolder as well.*

*-Woman, 92 years*

**Theme 2 – Motivation through enjoyment in the monitoring process and through physical and emotional changes**

This theme describes the pleasure users experience in monitoring (i.e., tracking physical activity, goal setting and the ability to see physical activity trends and goal achievement) their physical activity and how the monitoring may lead to motivation to be more physically active. Users could also experience physical- and emotional changes as they became more physically active while using the app.

The app could serve as a catalyst for motivation formation, encouraging users to engage in physical activity and the app's goal setting and tracking features were found to be a motivator for many, encouraging users to increase their physical activity levels.

*A goal was shown there, and that made it interesting to see week by week that I had walked as much as I set out to do, that I had the energy to do it.*

*-Woman, 77 years*

In increasing their physical activity levels, users could experience positive physical and psychological benefits. They experienced improved physical capacity, muscular strength, energy levels, and reduced symptom burden. User experiences in the emotional realm included both feelings of gratification and achievement by the goal setting and the tracking of the physical activity. Users explained how it felt rewarding to have performed some kind of activity and then get to track that activity in the app, and how that also instilled a sense of achievement. The app almost felt like a supporter or a coach, someone who would give you encouragement:

*It was kind of like a pat on the back, good boy!*

*-Man, 86 years*

They could also feel heightened emotional well-being and more confidence in increasing their physical activity levels. However, these experiences are nuanced by various motivators and barriers, especially due to health conditions or fluctuating motivation levels.

*If you have one of these and start doing [physical activity] four times a week, and then feel like 'hey, this went well', then the following week you increase it to six times. Eventually you are at 10 times instead of four. You see it, and it makes you feel good inside.*

*-Man, 74 years*

Most users expressed a willingness to continue using the app, because they believed it contributed positively to their health outcomes.

Conversely, individuals who currently feel well, may lack motivation to continue using the app, viewing it as unnecessary. Furthermore, those experiencing worsening health conditions may feel discouraged from even trying to use the app further, questioning its usefulness in managing their evolving health needs.

*I mean, I can't manage anything, I'm just getting worse and worse, so I think it's almost pointless.*

*-Woman, 83 years*

## Discussion

This study sought to describe the experiences of people with HF who used a physical activity support app (the Activity Coach) for 12 weeks. While there were some challenges associated with tracking physical activity, the app cultivated awareness of physical activity engagement. Users were motivated to increase their physical activity by setting goals and tracking their progress. Additionally, the physical and emotional changes they experienced further enhanced their motivation.

While longer follow-up times are essential for studying maintained behaviour change and clinical outcomes, new habits are usually considered being formed sooner than after 12 weeks [38]. Educating users about physical activity, setting physical activity goals, and tracking their physical activity enhanced their understanding of what constitutes physical activity and the potential benefits of increasing their activity levels. Becoming aware, is one of the first and crucial steps in behaviour change [39,40]. As users agreed to use the app for 12 weeks, they were in a receptive state for internalizing new knowledge about themselves and physical activity. Importantly, using the app made more aware of how sedentary they were or that they already performed a significant amount of physical activity. This increased awareness and acknowledgment of their physical activity levels, inspired users to change their physical activity. Since users reported both increased understanding of what would be appropriate physical activity for them and also increased physical activity, it seems that both declarative and procedural knowledge about physical activity was increased [41].

The app was designed to support physical activity, and not as an exercise tool. This means that it was to encourage doing daily physical activities and tracking these, and although the introduction week educated the user on what constitutes physical activity, some challenges in defining physical activity persisted. It was also difficult to remember what physical activity they had done earlier in the day, once they made the effort to track it. It is well known that recall of past physical activity is inversely correlated to the intensity of the physical activity [42], i.e., identifying, remembering and registering low intensity physical activity is difficult, and the users' experiences in this study confirmed that. Means to combat this might include visual or audible reminders, or possibly the incorporation of external sensors that either transfer registered activity into the app automatically or reminds the user to do so once a registration of physical activity has happened.

There were several aspects of the app that were experienced as being too rigid. Users wanted to be able to go back to previous days and add or change tracked data, they wished to be able to choose in what time increment physical activity was to be input and similarly they wished for more flexible goal setting. It should however be noted that any added features also adds complexity, and a review of mHealth as support for chronic conditions in the elderly found that a

minimally complex solution will be the most successful [43]. Others have also shown that inputting large amounts of data as opposed to simple tracking is frustrating [44] and that ease of use is one of the most commonly required features [45]. These sentiments were also observed in the qualitative parts of the development process of the app [23].

Forgetting to track physical activity appeared as an impediment to frequent use, suggesting some sort of prompt or cue might be appropriate. While some have published prompts or cues as crucial to success [44,46], others have described it as being perceived as nagging or annoying [47].

The experience of having performed some physical activity, and then getting to tracking it was rewarding and led to a sense of achievement. Setting goals and tracking their physical activity, was experienced as motivating for the users to increase their physical activity, which has also been reported by others in several reviews and is a commonly accepted strategy for promoting physical activity [48,49].

Being presented with goals to achieve, suggested by an mHealth app has previously been reported to induce negative stress in some [23], although no such experiences were found in this study.

Users of the app experienced improvements in physical parameters, which we assume is a distal effect of having achieved the proximal effect of increased physical activity, according to our theoretical model for the intervention [23]. There is some evidence to suggest that mHealth apps or tools can increase physical activity [50], but there is no evidence that clearly shows that mHealth-induced increased physical activity leads to improved clinical outcomes. Nevertheless, it is encouraging that the users experienced effects on mental health or well-being such as quality of life, which is an increasingly important outcome to strive for in the elderly, chronically ill community [51].

Furthermore, users could experience emotional changes, such as being more confidence to perform physical activity, i.e., increased self-efficacy. It is unclear whether the app enhanced self-efficacy first which then encouraged actual activities, or if it is the other way around but it is conceivable that the two positively influence each other. It has been shown that self-efficacy needs to be present before motivation will manifest as physical activity [52], but experiences in this study also suggest the other direction. The app's impact on supporting the users' health behaviours relating to physical activity, highlights the interplay between the sustained engagement, self-efficacy, and individual motivations.

Previous research has shown that exercise management is perceived as important [45] and that user trends and tailored goal setting in turn are important to achieve behaviour change relating to physical activity [43,53]. There is also generally an expectation that a tool would provide instruction and knowledge on how to correctly perform physical activity [54], as well as establishing an understanding of that the desired behaviour (i.e., physical activity in our case) leads to desirable outcomes [47]. The app incorporates explicit education and instructions on what physical activity is and can be for the intended population, and there is also a more subtle aspect of learning whereby interacting with the app, the user experiences improved understanding of the relationship between physical activity and health, which is likely to foster motivation for continued app use for health maintenance. The information and the users' own experiences of what the increased physical activity leads to, as well as access to trends and individually set goals, may have been key features in establishing the motivation to be more physically active, and seemingly also self-efficacy.

While the app was expected to be used primarily by older users, mHealth for cardiac disease has been identified as likely to be more successful among younger users [44]. However, there were indications in this study that age was not a determining factor but rather that the app would be more valuable to people with lower levels of physical activity or with a higher symptom burden, independent of age. The users who commented on that the app would be better suited for this group, were the ones classified as physically active, implying that the level of physical activity might predict user satisfaction with the app. Addressing diverse viewpoints is crucial for designing inclusive health technologies that accommodate the varying levels of health and motivation among users. If needs are not possible to meet, user selection might be the appropriate strategy to improve engagement in the technology in question. In future development or design work of interventions, it is advised to maintain (an update if needed) a theoretical framework, as theory-based interventions are more efficient in changing physical activity than interventions not based on a theory [55].

Since trends have been described to be important [43], and we could see that almost no users used that feature (the History tab), it should be investigated if some modifications could be done such that the trends were more visible or apparent in the daily use at the home screen, which could also be seen as a different take on one of the comments regarding an always present status gauge on the home page.

One could also speculate that increased interaction with a physiotherapist or health-care provider, could add value to the implementation of the app. Tracked activity could (possibly only on demand) be shared with the user's physiotherapist, who could then reply with supportive and encouraging text messages. It is important to consider stakeholders on several socio-ecological levels [21,56].

## Study design and limitations

Based on the pilot RCT it was only 10 users eligible for the interviews [30], it might have added value to have had a larger study population[57]. However, more users was not an option since it was limited by the number recruited in the intervention study [29]. In qualitative analysis, sample size considerations are used to ensure some degree of saturation [58], but the concept of saturation is not necessarily relevant for reflexive thematic analysis, where no assumptions on a "truth" to be found are made, and quality rather stems from depth of engagement with the data [59].

The users were a mix of physically active and physically inactive people, and as previously mentioned that resulted in somewhat different experiences. While it is possible that having all users belong to the same category would have made for a more homogenous group, it might be considered a strength of the study to include more diverse users.

All interviews were performed via mobile phone, which means that body language, facial expressions, and other non-verbal signals will not be recorded. However, telephone interviews are now considered valid and trustworthy alternatives to face-to-face interviews [60].

All the users in this study were previously already equipped with a home-based mHealth tool called Optilogg, on which the app was subsequently installed. This does introduce selection-bias in the study, as all users already had agreed to use mHealth, and thus negatively affects transferability.

## Conclusions

Users of the app experienced awareness of their physical activity, and positive physical and emotional changes. They felt motivated to increase their physical activity levels due to the enjoyment of monitoring their physical activity, however, challenges in defining and tracking physical activity were highlighted. Users suggested the app might be more beneficial for less active individuals or those with higher symptom burdens, indicating a need for screening. The study underscores the potential role of mHealth in increasing physical activity motivation and engagement in people with heart failure.

## Supporting information

**S1 File.** English_verbatim.
(ZIP)

## Author contributions

**Conceptualization:** Andreas Blomqvist, Anna Strömberg, Maria Bäck, Tiny Jaarsma, Leonie Klompstra.

**Data curation:** Andreas Blomqvist.

**Formal analysis:** Andreas Blomqvist, Marie Lundberg, Leonie Klompstra.

**Investigation:** Andreas Blomqvist.

**Methodology:** Andreas Blomqvist, Anna Strömberg, Marie Lundberg, Maria Bäck, Tiny Jaarsma, Leonie Klompstra.

**Project administration:** Andreas Blomqvist.

**Resources:** Tiny Jaarsma, Leonie Klompstra.

**Software:** Andreas Blomqvist.

**Supervision:** Anna Strömberg, Maria Bäck, Tiny Jaarsma, Leonie Klompstra.

**Validation:** Andreas Blomqvist, Tiny Jaarsma.

**Visualization:** Andreas Blomqvist.

**Writing – original draft:** Andreas Blomqvist.

**Writing – review & editing:** Andreas Blomqvist, Anna Strömberg, Maria Bäck, Tiny Jaarsma, Leonie Klompstra.

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
