## [Decision Letter · Decision Letter 0]

15 Oct 2024

PONE-D-24-34455Exploring User Experience: A Qualitative Analysis of the use of a Physical Activity Support App for People with Heart FailurePLOS ONE

Dear Dr. Blomqvist,

Thank you for submitting your manuscript to PLOS ONE. After careful consideration, we feel that it has merit but does not fully meet PLOS ONE’s publication criteria as it currently stands. Therefore, we invite you to submit a revised version of the manuscript that addresses the points raised during the review process.

We look forward to receiving your revised manuscript.

Kind regards,

Najmul Hasan, PhD

Academic Editor

PLOS ONE

2. Thank you for stating the following in the Competing Interests section: [I have read the journal's policy and the authors of this manuscript have the following competing interests:

The author Andreas Blomqvist, PhD-student at Linköping University, is an employee and shareholder of the company that developed the app and Optilogg (CareLigo AB, Sweden).]. Please confirm that this does not alter your adherence to all PLOS ONE policies on sharing data and materials, by including the following statement: "This does not alter our adherence to PLOS ONE policies on sharing data and materials.” (as detailed online in our guide for authors http://journals.plos.org/plosone/s/competing-interests). If there are restrictions on sharing of data and/or materials, please state these. Please note that we cannot proceed with consideration of your article until this information has been declared. Please include your updated Competing Interests statement in your cover letter; we will change the online submission form on your behalf.

3. In the online submission form, you indicated that [All data is available through request to the corresponding author.]. All PLOS journals now require all data underlying the findings described in their manuscript to be freely available to other researchers, either 1. In a public repository, 2. Within the manuscript itself, or 3. Uploaded as supplementary information. This policy applies to all data except where public deposition would breach compliance with the protocol approved by your research ethics board. If your data cannot be made publicly available for ethical or legal reasons (e.g., public availability would compromise patient privacy), please explain your reasons on resubmission and your exemption request will be escalated for approval.

Additional Editor Comments (if provided):

Reviewers' comments:

Reviewer's Responses to Questions

**Comments to the Author**

1. Is the manuscript technically sound, and do the data support the conclusions?

Reviewer #1: Partly

Reviewer #2: Partly

2. Has the statistical analysis been performed appropriately and rigorously? 

Reviewer #1: N/A

Reviewer #2: No

3. Have the authors made all data underlying the findings in their manuscript fully available?

Reviewer #1: Yes

Reviewer #2: No

4. Is the manuscript presented in an intelligible fashion and written in standard English?

Reviewer #1: Yes

Reviewer #2: Yes

5. Review Comments to the Author

Reviewer #1: Hello,

Thanks for your submission on this manuscript titled "Exploring User Experience: A Qualitative Analysis of the use of a Physical Activity Support App for People with Heart Failure" to this journal. I have following suggestions and seek clarifications -

1. The sample size of the study is very small (10) and this study should have been just a proof of concept while many such studies exist for various other apps . can you please explain why the sample size was so small ? Please mention the same as limitation of the study.

2. Were the questions ever validated in past ?

3. What was the median/mean duration of each question answered and time from which the interview was done post use of app ?

4. Please elaborate more on the app technicalities

Thanks

Reviewer #2: Thank you for the opportunity to review this manuscript, which addresses an important and timely issue regarding the use of mobile health (mHealth) technologies to promote physical activity in patients with heart failure. While this is an important topic with potential public health benefits, there are major shortcomings that need to be addressed, including study design, lack of technological innovation, methodological limitations, insufficient depth of analysis, and overall rigor of the study. Below, I outline these concerns in detail.

Major Concerns:

1. This study presents interesting results, but is limited by its small sample size (n=10). The statistical methods used are not sufficient to support the study conclusions, which weakens the generalizability and reliability of the results. More rigorous statistical techniques and a larger dataset are necessary to validate the results. Furthermore, while thematic analysis is appropriate for qualitative research, this manuscript lacks depth in discussing the practical implications of the results in a broader clinical context.

2. Additionally, the app under study lacks technological innovation. While the app is a step in the right direction, the current version does not offer any unique features compared to existing fitness tracking apps and therefore does not demonstrate significant innovation. There is no explanation to the connection between this app and heart failure.

3. There are contradictions within the results section. For instance, while Line 190-191 indicate that users experienced challenges in tracking physical activities, Line 199 suggests that many users found the app easy to navigate. This contradiction raises concerns about the data's reliability and the manuscript's internal consistency, and these contradictions and their implications for real-world use are not sufficiently addressed.

4. The study focuses on short-term use (12 weeks), but long-term behavior change is crucial for managing heart failure. A longer follow-up period or more discussion on long-term engagement is needed.

5. The manuscript lacks an in-depth usability testing framework, which is essential for evaluating the app's effectiveness, especially given the target audience of elderly users. Additionally, the study lacks more in-depth analysis of how the app is targeted to attract user behaviors through UX terminology and methodologies. The overall user experience is spoken at too general of a level and fails to tap into describing key users’ tasks that are critical in describing the entire experience.

Minor:

1. There is a potential conflict of interest because one of the authors is an employee and shareholder of the company that developed the app being evaluated. This relationship was disclosed, but there is insufficient information about how the study mitigated bias through independent assessment or objective safeguards regarding the study design, data collection, or interpretation of results.

2. Figure 1 should be improved in quality, and Table 2 would benefit from additional data points, such as age, gender, and symptom classification, to provide more context to the analysis.

3. There is inconsistency regarding the target audience In regard to age group, the introduction emphasizes the elderly, but Line 356 & 358 refer to "younger users," causing confusion about the app's intended user base. The study initially indicated that the focus for the app as well is for those with HF, but in the findings indicated that users additionally had other conditions, which could have also influenced the user’s behaviors in relation to using the app.

4. The study also shows concerns about the transparency of the data, not only because of the weak quantitative analysis, but also because of the questionable availability of the raw data according to the journal's policy.

5. The manuscript contains several grammatical errors and unclear sections. For example, vague terms such as “symptoms” need to be clarified, and more clarity is needed in the introduction, particularly in linking the purpose of the study to the gaps in existing research

6. Line 61: Unclear how “portable” address to line 54-55 “comorbidities, … and transportation”

7. Line 95-96: “if the user…. About 75-100 words” � Does this really matter?

8. Line 182: “Cultivating awareness…” � Does this refer to awareness of an individual and their physical activity.

9. Line 190-191: “users did experience some challenges in defining and effectively tracking their physical activities during the daily use” is contradiction to Line 199 “many users found the app easy to navigate and use”

10. Line 268-273: Who is the target audience for the app?

11. Line 356 & 358: “younger users” & “ independent of age” � In introduction mentions usability of elderly

In conclusion, while the manuscript addresses an important area of mHealth applications for individuals with heart failure, the manuscript requires significant revisions in methodology, innovation, and clarity to meet the standards for publication. I encourage the authors to address these concerns and resubmit, as I believe this study has the potential to make a meaningful contribution to the field.

6. PLOS authors have the option to publish the peer review history of their article (what does this mean? ). If published, this will include your full peer review and any attached files.

**Do you want your identity to be public for this peer review?** For information about this choice, including consent withdrawal, please see our Privacy Policy .

Reviewer #1: **Yes: ** DR KAMAL H SHARMA

Reviewer #2: No

---

## [Author Response · Author response to Decision Letter 1]

2 Dec 2024

Reviewer #1: Hello,

Thanks for your submission on this manuscript titled "Exploring User Experience: A Qualitative Analysis of the use of a Physical Activity Support App for People with Heart Failure" to this journal. I have following suggestions and seek clarifications -

1. The sample size of the study is very small (10) and this study should have been just a proof of concept while many such studies exist for various other apps . can you please explain why the sample size was so small ? Please mention the same as limitation of the study.

-Under “Ethical approval and population” we describe the reason for the small sample size. A pilot-RCT was performed using the intervention (“the activity coach”). All study participants from that pilot-RCT who were randomized to the intervention arm were approached to participate in this study, and all who conceded were included.

-Under “Study design and limitations” we write that “it might have added value to have had a larger study population”. We have however, as suggested by the reviewer, also elaborated on sample size in the context of reflexive thematic analysis in this Limitations-section.

2. Were the questions ever validated in past ?

-The questions in the interview guide were created based on other published interview guides from similar contexts (Anderson et al. Mobile health apps to facilitate self-care. PloS one. 2016; and Schueller et al. Understanding people’s use of and perspectives on mood-tracking apps. JMIR mental health. 2021)

3. What was the median/mean duration of each question answered and time from which the interview was done post use of app ?

-This information has been added under “Data collection”.

4. Please elaborate more on the app technicalities

-Thanks

This has been elaborated on under “The mHealth tool”.

Reviewer #2: Thank you for the opportunity to review this manuscript, which addresses an important and timely issue regarding the use of mobile health (mHealth) technologies to promote physical activity in patients with heart failure. While this is an important topic with potential public health benefits, there are major shortcomings that need to be addressed, including study design, lack of technological innovation, methodological limitations, insufficient depth of analysis, and overall rigor of the study. Below, I outline these concerns in detail.

Major Concerns:

1. This study presents interesting results, but is limited by its small sample size (n=10). The statistical methods used are not sufficient to support the study conclusions, which weakens the generalizability and reliability of the results. More rigorous statistical techniques and a larger dataset are necessary to validate the results. Furthermore, while thematic analysis is appropriate for qualitative research, this manuscript lacks depth in discussing the practical implications of the results in a broader clinical context.

-Thank you. In this paper we used a qualitative method, aimed to describe the experiences in a qualitative way, where the focus is not on statistics. We used reflexive thematic analysis, where statistical considerations are not appropriate but rather the narratives of the users are important.

-With regards to the sample size, we have elaborated the discussion on the topic under “Study design and limitations”.

2. Additionally, the app under study lacks technological innovation. While the app is a step in the right direction, the current version does not offer any unique features compared to existing fitness tracking apps and therefore does not demonstrate significant innovation. There is no explanation to the connection between this app and heart failure.

-The Activity Coach specifically addresses so called incidental or non-exercise physical activity (NEPA), which is different than exercise (e.g. used in “fitness tracking apps”). This NEPA affects prognosis in the heart failure population. While the body of science on exercise interventions for the heart failure population is extensive, there is to our knowledge no published science on mHealth interventions specifically targeting NEPA in the HF community. As such, this is a rather unique intervention.

The authors thus maintain that this is a poorly studied field, and one that is very important as the heart failure population need support in NEPA according to leading bodies on cardiovascular prevention and rehabilitation.

-We have added some clarifying remarks regarding this in the introduction.

3. There are contradictions within the results section. For instance, while Line 190-191 indicate that users experienced challenges in tracking physical activities, Line 199 suggests that many users found the app easy to navigate. This contradiction raises concerns about the data's reliability and the manuscript's internal consistency, and these contradictions and their implications for real-world use are not sufficiently addressed.

-We understand this confusion. The challenge was in “defining physical activity”, not the concrete tracking of it in the app. We have clarified this under “Theme 1 – Cultivating awareness of physical activity engagement”.

4. The study focuses on short-term use (12 weeks), but long-term behavior change is crucial for managing heart failure. A longer follow-up period or more discussion on long-term engagement is needed.

-The authors agree that long-term data would be valuable, and we have added this in the discussion.

5. The manuscript lacks an in-depth usability testing framework, which is essential for evaluating the app's effectiveness, especially given the target audience of elderly users. Additionally, the study lacks more in-depth analysis of how the app is targeted to attract user behaviors through UX terminology and methodologies. The overall user experience is spoken at too general of a level and fails to tap into describing key users’ tasks that are critical in describing the entire experience.

-The aim of the study was to “describe users’ experiences”, not to assess usability nor to describe underlying theory. Both usability and aspects of how behaviours are targeted has been studied and published previously (Blomqvist, 2024, BMC Medical Informatics and Decision Making, Usability and feasibility analysis of an mHealth-tool).

Minor:

1. There is a potential conflict of interest because one of the authors is an employee and shareholder of the company that developed the app being evaluated. This relationship was disclosed, but there is insufficient information about how the study mitigated bias through independent assessment or objective safeguards regarding the study design, data collection, or interpretation of results.

-Thank you for pointing this out to us and the opportunity to clarify this. We have added some information as to how this potential bias was mitigated, under “Data collection”.

2. Figure 1 should be improved in quality, and Table 2 would benefit from additional data points, such as age, gender, and symptom classification, to provide more context to the analysis.

-We have updated the figure with higher resolution.

-Table 2 contains age and gender and NYHA classification is a classification of the levels of heart failure related symptoms.

3. There is inconsistency regarding the target audience In regard to age group, the introduction emphasizes the elderly, but Line 356 & 358 refer to "younger users," causing confusion about the app's intended user base. The study initially indicated that the focus for the app as well is for those with HF, but in the findings indicated that users additionally had other conditions, which could have also influenced the user’s behaviors in relation to using the app.

-Thank you, we have corrected this. Regarding other conditions, we listed the baseline demographics in Table 2, so the reader can have a better understanding of the health status of the studied population.

4. The study also shows concerns about the transparency of the data, not only because of the weak quantitative analysis, but also because of the questionable availability of the raw data according to the journal's policy.

-A strictly qualitative methodology was used to ensure trustworthiness. We will make sure to clarify that all data in the form of verbatim transcripts are available, according to the journal’s policy.

5. The manuscript contains several grammatical errors and unclear sections. For example, vague terms such as “symptoms” need to be clarified, and more clarity is needed in the introduction, particularly in linking the purpose of the study to the gaps in existing research

-Thank you, we have gone through the manuscript and have tried to improve on grammar and language clarity.

6. Line 61: Unclear how “portable” address to line 54-55 “comorbidities, … and transportation”

-Thank you, this has been clarified.

7. Line 95-96: “if the user…. About 75-100 words” � Does this really matter?

-Thank you for this comment. While developing the app, we found published literature on mHealth adoption and usability, which indicate that finding a balance between ease of use (i.e. not too “cluttered” appearance) and easily accessed information is important. This was mentioned to clarify how that was achieved, although we concede it might be superfluous.

8. Line 182: “Cultivating awareness…” � Does this refer to awareness of an individual and their physical activity.

-Yes. We have clarified in “Theme 1 – Cultivating awareness of physical activity engagement”, where we describe what that theme means.

9. Line 190-191: “users did experience some challenges in defining and effectively tracking their physical activities during the daily use” is contradiction to Line 199 “many users found the app easy to navigate and use”

-We clarified this. Please see the answer to “Major concerns #3” for a more elaborate response.

10. Line 268-273: Who is the target audience for the app?

-The mHealth-tool designed for supporting physical activity in people with HF, which implicitly means that the majority of users will likely be elderly. While we under “Discussion” discuss whether it might be more suitable for people who are physically inactive, the current manuscript did not employ such criteria.

11. Line 356 & 358: “younger users” & “ independent of age” � In introduction mentions usability of elderly

We believe that the most likely user of the app will be elderly, and as a consequence usability for elderly users was believed to be important.

-In conclusion, while the manuscript addresses an important area of mHealth applications for individuals with heart failure, the manuscript requires significant revisions in methodology, innovation, and clarity to meet the standards for publication. I encourage the authors to address these concerns and resubmit, as I believe this study has the potential to make a meaningful contribution to the field.

-We thank the reviewer for constructive comments. We maintain that the methodology being strictly qualitative is sound, and not methodologically flawed by not being quantitative in nature, as suggested. We also thank the reviewer for the encouraging concluding remark.

---

## [Decision Letter · Decision Letter 1]

9 Feb 2025

Exploring User Experience: A Qualitative Analysis of the use of a Physical Activity Support App for People with Heart Failure

PONE-D-24-34455R1

Dear Dr. Blomqvist, 

We’re pleased to inform you that your manuscript has been judged scientifically suitable for publication and will be formally accepted for publication once it meets all outstanding technical requirements.

Kind regards,

Najmul Hasan, PhD

Academic Editor

PLOS ONE

Reviewers' comments:

Reviewer's Responses to Questions

**Comments to the Author**

1. If the authors have adequately addressed your comments raised in a previous round of review and you feel that this manuscript is now acceptable for publication, you may indicate that here to bypass the “Comments to the Author” section, enter your conflict of interest statement in the “Confidential to Editor” section, and submit your "Accept" recommendation.

Reviewer #2: All comments have been addressed

2. Is the manuscript technically sound, and do the data support the conclusions?

Reviewer #2: Yes

3. Has the statistical analysis been performed appropriately and rigorously? 

Reviewer #2: N/A

4. Have the authors made all data underlying the findings in their manuscript fully available?

Reviewer #2: Yes

5. Is the manuscript presented in an intelligible fashion and written in standard English?

Reviewer #2: Yes

6. Review Comments to the Author

Reviewer #2: The authors are responsive to most of the reviewers' comments and appear to have made significant efforts to improve the manuscript, so I recommend acceptance of the manuscript. I note that appropriate revisions have been made throughout the manuscript to improve the clarity and coherence of the manuscript, including clarification of the qualitative research methodology, concerns about sample size, and further elaboration on the technical aspects of the app and its relevance to people with heart failure. In particular, the explanation of why the app focuses on physical activity other than exercise (NEPA) and why this is important for people with heart failure is good. The authors have effectively explained their methodological choices and demonstrated a commitment to scientific rigor by acknowledging the limitations of the study. The revised manuscript provides valuable insights into the user experience of an mHealth tool to support physical activity in people with heart failure. It has the potential to make a scientific contribution at a time of increasing interest in mHealth technologies for chronic disease management, and I therefore recommend its acceptance.

7. PLOS authors have the option to publish the peer review history of their article (what does this mean? ). If published, this will include your full peer review and any attached files.

**Do you want your identity to be public for this peer review?** For information about this choice, including consent withdrawal, please see our Privacy Policy .

Reviewer #2: No

---

## [Editor Report · Acceptance letter]

PONE-D-24-34455R1

PLOS ONE

Dear Dr. Blomqvist,

I'm pleased to inform you that your manuscript has been deemed suitable for publication in PLOS ONE. Congratulations! Your manuscript is now being handed over to our production team.

Kind regards,

on behalf of

Dr. Najmul Hasan

Academic Editor

PLOS ONE